# What Do We Know about the Microbiome in Cystic Fibrosis? Is There a Role for Probiotics and Prebiotics?

**DOI:** 10.3390/nu14030480

**Published:** 2022-01-22

**Authors:** Josie M. van Dorst, Rachel Y. Tam, Chee Y. Ooi

**Affiliations:** 1Discipline of Paediatrics & Child Health, Randwick Clinical Campus, School of Clinical Medicine, UNSW Medicine & Health, UNSW, Sydney 2031, Australia; j.vandorst@unsw.edu.au (J.M.v.D.); yantungrachel.tam@unsw.edu.au (R.Y.T.); 2Molecular and Integrative Cystic Fibrosis (miCF) Research Centre, Sydney 2031, Australia; 3Department of Gastroenterology, Sydney Children’s Hospital Randwick, Sydney 2031, Australia

**Keywords:** cystic fibrosis, dysbiosis, inflammation, nutrition, prebiotic, probiotic

## Abstract

Cystic fibrosis (CF) is a life-shortening genetic disorder that affects the cystic fibrosis transmembrane conductance regulator (CFTR) protein. In the gastrointestinal (GI) tract, CFTR dysfunction results in low intestinal pH, thick and inspissated mucus, a lack of endogenous pancreatic enzymes, and reduced motility. These mechanisms, combined with antibiotic therapies, drive GI inflammation and significant alteration of the GI microbiota (dysbiosis). Dysbiosis and inflammation are key factors in systemic inflammation and GI complications including malignancy. The following review examines the potential for probiotic and prebiotic therapies to provide clinical benefits through modulation of the microbiome. Evidence from randomised control trials suggest probiotics are likely to improve GI inflammation and reduce the incidence of CF pulmonary exacerbations. However, the highly variable, low-quality data is a barrier to the implementation of probiotics into routine CF care. Epidemiological studies and clinical trials support the potential of dietary fibre and prebiotic supplements to beneficially modulate the microbiome in gastrointestinal conditions. To date, limited evidence is available on their safety and efficacy in CF. Variable responses to probiotics and prebiotics highlight the need for personalised approaches that consider an individual’s underlying microbiota, diet, and existing medications against the backdrop of the complex nutritional needs in CF.

## 1. Introduction

Cystic fibrosis (CF) is a genetic condition of autosomal recessive inheritance related to mutations in the gene coding for the cystic fibrosis transmembrane conductance regulator (CFTR) protein [1]. The CFTR protein affects the fluid secretion and mucus hydration of epithelial cells in the airway, pancreas, intestines, and hepatobiliary tracts [2]. Chronic suppurative respiratory disease arising due to impaired clearance of dehydrated airway secretions is typically the principal cause of morbidity and mortality. However, the majority (>90%) of patients with CF also suffer from gastrointestinal (GI) symptoms and complications [3,4]. Dysfunction of the CFTR protein in the GI system results in low intestinal pH, thick and inspissated mucus, a lack of endogenous pancreatic enzymes, reduced motility, and possibly an impaired innate immunity [5,6,7] (Figure 1). These mechanisms are proposed drivers of local GI inflammation and contribute to a range of intestinal morbidities, including an increased risk of early-onset adult GI cancer [8,9,10,11]. GI dysfunction combined with antibiotic therapies also drives significant alteration (dysbiosis) of the GI microbiota (Figure 1). Altered CF microbiota is likely to compound the proinflammatory effects of the underlying disease.

Evidence is accumulating that GI bacterial strains, which occur differentially between CF and healthy controls (HC), are linked to inflammatory [12,13,14] and malignancy processes [7,14]. Supplementation with prebiotics and probiotics are thought to provide clinical benefit by promoting commensal bacteria and biosynthesis of immunomodulatory metabolites. As public awareness and acceptance of probiotics and prebiotics continue to expand, there is a growing interest in the potential clinical benefits of dietary prebiotics and probiotics in CF. A total of 17 probiotic trials, including 12 RCTs, have thus far investigated the safety and efficacy of individual probiotic strains and strain combinations in children and adults with CF [15]. Promising improvements in inflammation [16,17,18], nutritional status [19], and health outcomes [20,21] have been observed. However, due to selective reporting and incomplete outcome data, the certainty of evidence has been evaluated as low. Furthermore, large variations between protocols, probiotic formulas, dosage, and duration of treatments limit the potential for clinical application. Prebiotics are often included in probiotic preparations, but evidence surrounding safety and efficacy for the exclusive use of prebiotics is trailing. There is only one clinical trial investigating the prebiotic high-amylose maize starch (HAMS) in adults with CF [22]. The efficacy of prebiotic supplementation is based on the selective utilisationof substrates (usually indigestible carbohydrates) by beneficial bacteria. Critically, it is not yet known whether the altered CF intestinal microbiota retains the capacity to exploit prebiotic substrates.

This review describes the physiology of the GI tract in CF and the clinical relevance of GI microbiome dysbiosis and inflammation. We discuss the current understanding of probiotic and prebiotic mechanisms of action, provide important examples of clinical studies examining probiotic and prebiotic applications in CF, and discuss considerations for clinical translation.

## 2. Cystic Fibrosis in the Gastrointestinal Tract

CFTR is an important contributor to the normal physiology of the gastrointestinal (GI) tract; as such, its dysfunction in CF disease has profound impacts on GI homeostasis. CFTR is an epithelial cyclic adenosine monophosphate (cAMP)-dependent anion-selective channel. It primarily secretes bicarbonate and chloride, and therefore exerts great influence on the acidity and viscosity of secretions. In CF, the dysfunction of this ion channel is clearly manifested in the systemic production of hyperacidic and viscid mucus [23,24]. CFTR also plays a role in the maintenance of epithelial tight junctions, modulation of fluid flow, regulation of ion channels (such as sodium, potassium, calcium, and other chloride channels [25,26]), and coordination of gut motility [27,28]. Altogether, disruptions to these normal and vital functions of CFTR culminate in an abnormal GI tract (Figure 1).

The altered GI environment in CF results in various clinical sequelae that can be collectively referred to as “obstructive tubulopathies.” The most common of these is pancreatic insufficiency, which affects as much as 90% of patients with CF [29]. From as early as in utero, the presence of concentrated pancreatic ductal secretions leads to luminal protein precipitation with resultant obstruction and dilation of the pancreatic ducts, culminating in progressive, irreversible destruction and fibrosis of the acinar tissue. The resultant pancreas is dysfunctional and severely impaired in its ability to secrete critical enzymes necessary for the digestion of carbohydrates, fats, and proteins [30,31]. Obstructive tubulopathies are also evident in the intestines in the form of meconium ileus (MI) and distal intestinal obstruction syndrome (DIOS). There are numerous other GI manifestations of CF, including gastroesophageal reflux disease, pancreatitis, and liver disease, which have been detailed elsewhere [32,33,34,35,36]. There are also less clinically obvious, but equally significant, manifestations that arise as a result of the altered GI milieu in CF; namely, alterations to the gut microbiota and intestinal inflammation. These are discussed in detail below.

## 3. The Human Gut Microbiome

The gut microbiome is a sophisticated, functional environment comprising an abundance of microbes along the GI tract. These microorganisms and their metabolites perform homeostatic functions, including the regulation of the gastrointestinal epithelial barrier, fermentation of dietary starches and fibres, synthesis of amino acids and essential vitamins, and modulation of the immune system locally and distally [37,38]. While a small proportion of the gut microbiota is heritable, it is largely influenced by nongenetic factors [39]. The early development of the gut microbiome in infancy is predominantly shaped by one’s mode of birth and feeding, with the cessation of breastfeeding being the driver of functional maturation into an adultlike microbiota [40,41]. Subsequently, diet plays a primary role in shaping the gut microbiome, as organisms respond to selective pressures from dietary patterns throughout life [42,43]. Numerous other environmental factors can also affect the gut microbiota, but perhaps the most well-established are medications, including antibiotics [41,44,45,46].

One important aspect to spotlight when characterising the gut microbiota is microbial diversity. Microbial diversity refers to species richness (the number of species) and/or evenness (the relative distribution of species). Reduced microbial diversity is broadly associated with ill health, as it is hypothesised that species diversity confers the ability to withstand environmental threats and maintain homeostasis. This is attributable to compensatory functional redundancies enabled by a more robust ecological environment [47,48]. Disruption to the normal composition, physiology, and diversity of the gut microbiota is an increasingly recognised feature of numerous disease processes. Decreased microbial diversity has repeatedly been observed in patients with chronic conditions, including obesity, inflammatory bowel disease (IBD), type 1 and type 2 diabetes mellitus, and asthma. Dysbiosis, the collective term for alterations to the normal balance or composition of gut microbes, is also evident in many of those disease processes [48,49,50,51]. These observations point to the critical involvement of the gut microbiota in health and disease, and solidify the rationale for utilising microbial modulation as a therapeutic target.

## 4. The CF Gut Microbiome

Given the significant alterations to the intestinal environment resulting from CFTR dysfunction, it is unsurprising that the CF gut microbiome differs from that of the healthy gut from early life onwards. One key features of the CF gut microbiome is decreased species diversity [7,12,52,53,54] (Figure 1). In addition, paediatric studies have demonstrated that the CF gut microbiome diversifies and matures at a significantly slower rate than that of a healthy child [54,55,56]. Compositionally, the CF gut microbiome also differs from that of the healthy gut. Reductions in *Bacteroidetes, Ruminococcaceae, Bifidobacterium,* and *Roseburia* have consistently been observed. In contrast, abundances of *Enterococcus, Veillonella*, and *Enterobacter* have been shown to be relatively increased in the CF gut [7,12,52,53,54,55,57]. Use of the CFTR modulator ivacaftor is associated with arguably “healthier’’ microbiome profiles, reinforcing the concept that dysbiosis is driven by CFTR dysfunction [58]. Recent advancements in metagenomic methods have enhanced the ability to characterise the functionality of the gut microbiota, thereby elucidating the physiological consequences of dysbiosis. It has been demonstrated that the CF gut microbiome displays an increased capacity to metabolise nutrients, antioxidants, and short-chain fatty acids (SCFAs), as well as a relatively decreased propensity to synthesise fatty acids [7,56,59].

The key drivers of these changes to the gut microbiota involve the downstream effects of CFTR dysfunction. The production of dehydrated mucus, changes to intestinal pH, nutrient malabsorption, and prolonged intestinal transit secondary to intestinal dysmotility all have the potential to exert selective pressure on enteric microorganisms and ultimately alter the microbiome [60,61,62]. Notably, fat malabsorption following exocrine pancreatic insufficiency could also confer survival advantage to certain organisms that adapt well to high-fat intestinal environments [63]. These CFTR-related factors are further compounded by iatrogenic causes. Antibiotic exposure, which is prevalent in CF for the prophylaxis and treatment of respiratory tract infections, may contribute to changes in the gut microbiota. Studies in the CF population have consistently demonstrated an association between antibiotic use and decreased alpha diversity (within-sample species diversity) in the gut [12,53,64,65] (Figure 1). Multiple studies have also highlighted a correlation between antibiotic exposure and relative depletions of the bacterial genus *Bifidobacterium* [64,65,66,67]. The high-energy and high-fat diet prescribed in CF is another likely contributor (discussed below).

## 5. Intestinal Inflammation

Disruption to the gut microbiota is associated with intestinal inflammation in CF. Chronic inflammation is a well-recognised feature of the CF intestine, primarily evidenced by elevated faecal inflammatory markers in patients with CF in many studies [68,69,70,71,72,73] (Figure 1). The earliest evidence of GI inflammation was elevated concentrations of inflammatory markers such as interleukin-8, interleukin-1β, neutrophil elastase, and immunoglobulins on whole-gut lavage, reported by Smyth et al. [74]. Imaging techniques including endoscopy and capsule endoscopy have subsequently revealed a high prevalence of mucosal pathologies, including ulcerations and oedema in the CF GI tract [71,75,76].

Gut inflammation in CF is of a multifactorial aetiology. Mucus hyperviscosity and hyperacidity as a result of CFTR dysfunction likely promote gut inflammation [8,77,78]. CFTR itself is also involved in downregulating proinflammatory pathways, and hence its dysfunction in CF may contribute to the altered intestinal milieu [79] (Figure 1). Additionally, inflammation may be precipitated by intestinal dysmotility and the intraluminal pooling of inspissated contents [77,80]. The same iatrogenic factors that contribute to intestinal dysbiosis, namely antibiotic exposure and the high-fat CF diet, have also been shown to be correlated with intestinal inflammation in CF and other contexts [81,82,83] (Figure 1). The mechanisms by which antibiotics may induce inflammation are not well-known. However, it has been demonstrated in animal models that antibiotic administration promotes the translocation of microorganisms through goblet-cell-mediated pathways, subsequently increasing the release of inflammatory cytokines [84].

Notably, the aforementioned dysbiosis is a key contributor to intestinal inflammation in CF. Reductions in the abundances of bacteria with anti-inflammatory properties, including *Faecalibacterium prausnitzii,* has been widely observed in CF cohorts [7,52,53,66,70,85]. Many of these bacteria are known producers of short-chain fatty acids (SCFAs), the primary metabolites of anaerobic fermentation of dietary fibres and starches. SCFAs perform homeostatic functions, including intestinal epithelial maintenance, colonocyte nourishment, and immunomodulation (Figure 1). Accordingly, a relative depletion of SCFA-producing bacteria and subsequent reductions in SCFA levels may contribute to inflammation [60,86,87]. This is supported by numerous animal models in which SCFAs have been shown to improve epithelial integrity and ameliorate intestinal inflammation [88,89,90,91,92,93]. However, the interactions between the microbiota and inflammation are also bidirectional. Chronic inflammation results in the release of reactive oxygen and nitrogen species that supply terminal electron acceptors required for anaerobic respiration. This exerts selective pressure on gut microbes and may contribute to dysbiosis, as organisms with the ability to efficiently perform anaerobic respiration have a growth advantage [94]. For example, intestinal inflammation is associated with the proliferation of *Enterobacteriaceae*, a bacterial family that has high nitrate reductase activity and can undergo efficient nitrate respiration [95,96]. Indeed, organisms within the *Enterobacteriaceae* family (i.e., the *Enterobacter* genus) are relatively more abundant in the CF gut [7,12,64,85]. Many of the mechanistic aspects of the relationship between the gut microbiota and inflammation remain unknown, highlighting the intricacy of these complex interactions.

## 6. Nutritional Management in CF

In 1988, Corey et al. [97] published a landmark study that led to pivotal paradigm shifts in CF nutritional optimisation. It has since been established that energy requirements are increased in CF due to increased energy expenditure from chronic lung inflammation and increased work of breathing, as well as malabsorption secondary to exocrine pancreatic insufficiency and gastrointestinal disease [29,98,99]. Patients with CF are also at risk of deficiencies in fat-soluble vitamins due to fat and bile acid malabsorption, which often necessitates supplementation [100,101]. Good nutritional status beginning in childhood is now well-documented to be associated with better pulmonary function and survival in CF [102,103,104,105,106]. Body mass index (BMI) is positively correlated with forced expiratory volume in 1 second (FEV1) [107,108,109,110], and a low BMI at the age of 10 years is a risk factor for lung transplantation in adulthood [111]. Greater weight-for-age percentile at the age of 4 years is also associated with better pulmonary function and survival through to 18 years, as well as a reduced likelihood of subsequent pulmonary exacerbations, hospitalisations, or CF-related diabetes [106].

Today, patients with CF are recommended a high-energy diet (110–200% of the age- and sex-appropriate recommended daily energy intake) to maintain growth. While macronutrient targets are individual-specific, the current consensus generally advises that 15–20% of total energy intake be derived from protein, 40–45% from carbohydrates, and up to 35–40% from fat [29,100,112]. Despite the clear benefits of nutritional optimisation, it has become increasingly evident that patients with CF tend to overconsume “energy-dense, nutrient-poor” foods high in salt, sugar, and saturated fat (i.e., junk foods) in order to meet daily macronutrient requirements [113,114,115,116,117]. The proportion of patients with CF who are overweight or obese is increasing. While patients who are overweight or obese are reported to have better lung function than their normal weight or underweight counterparts in some studies, this finding may be confounded by the fact that these patients are also more likely to be pancreatic sufficient and have milder disease genotypes [118,119,120,121,122]. Additionally, weight gain and increased BMI, fat mass, and fat-free mass are reported outcomes of CFTR modulator therapies that need to be taken into account as modulator therapies gradually become the cornerstone of CF treatment [123,124,125,126]. Importantly, both high-fat diets and obesity may exacerbate existing alterations in gut microbial composition and chronic intestinal inflammation, with important clinical implications for individuals with CF [81,82,83,127,128] (Figure 1).

## 7. Clinical Significance of the CF Gut Microbiome

Intestinal dysbiosis and inflammation have been demonstrated to be significantly associated with clinical outcomes. In a recent study, Hayden et al. [55] observed a distinctly more marked dysbiosis in infants with CF who had low length compared to infants with CF who had normal length. Notably, the gut microbiome of infants with low length exhibited a reduced abundance of *Bacteroidetes* and relatively delayed maturation compared to that of infants with normal length [55]. Coffey et al. [7] had also previously reported a positive correlation between *Ruminococcaceae UCG 014* and BMI. Additionally, the CF intestinal microbiota contains a comparatively lower prevalence of proteins that facilitate carbohydrate transport, metabolism, and conversion, which may impact nutrient utilisation and thus adversely affect growth [85]. It has also been demonstrated that faecal calprotectin levels are inversely correlated with weight and height z-scores, and elevated calprotectin levels are associated with underweight BMI (<18.5 kg/m^2^) [13,70,72].

Additionally, there is a growing body of evidence that suggests that the intestinal microbiome is related to lung function. It has been reported that patients with lower FEV1 have reduced intestinal microbial diversity compared to their counterparts with better pulmonary function [12]. Positive correlations between intestinal bacterial genera such as *Ruminococcaceae NK4A214* and FEV1 have also previously been documented [7]. Furthermore, one study reported an association between microbial diversity in the gut microbiota and pulmonary exacerbation events [57]. Some studies have also demonstrated associations between gut inflammation and lower FEV1, although this has not yet been widely validated [68,72]. It is postulated that these associations reflect a physiological phenomenon termed the “gut–lung axis.” Along this axis, the intestinal and respiratory microbiota engage in cross-talk to regulate immunity and homeostasis in both the enteric and pulmonary environments [129]. In the intestinal compartment, this is achieved by gut-microbiota-derived metabolites, including SCFAs, which coordinate immune cell signaling cascades that ultimately involve the lungs through G-protein coupled receptor (GPCR)-mediated pathways and histone deacetylase inhibition [130,131,132]. In support of this, Hoen et al. [133] observed in a paediatric CF cohort that pulmonary colonisation with the pathogen *Pseudomonas aeruginosa*, a known contributor to declining lung function, was preceded by a reduction in the abundance of *Parabacteroides* in the gut. Notably, *Parabacteroides* is associated with immunomodulation and anti-inflammatory properties [134]. While there remain many unknowns with regard to the mechanistic aspects of the gut–lung axis, these findings suggested that the intestinal microbiome is a site of therapeutic potential that could be manipulated to optimise lung function.

Intestinal dysbiosis and inflammation have been linked to a number of serious morbidities. Firstly, while patients with CF do not typically present with overt GI symptoms similar to those of inflammatory bowel disease (IBD), elevated faecal calprotectin levels are correlated with a worse quality of life [4,135,136]. Elevated calprotectin has also been highlighted as a predictive factor of GI-related hospitalisations for infants with CF in their first year of life [136,137]. Importantly, intestinal dysbiosis and inflammation may be contributors to the increased risk of GI malignancies that is evident in the CF population [138]. While a clear causative mechanism has yet to be established in the context of CF, it is well recognised from studies pertaining to IBD that chronic inflammation poses a significant risk for the development of GI cancers [139,140]. This is largely due to oxidative stress and the resultant DNA damage, culminating in epigenetic disturbances to the expression of tumour-suppressive regulatory proteins, transcription factors, and signalling molecules [141,142]. Furthermore, the inflamed gut may confer a growth advantage to genotoxic organisms, especially *E. coli* [143]. Indeed, the relative abundance of *E. coli* is increased in CF, as well as in IBD and colorectal cancer [144,145]. The depletion of SCFA-producing organisms in the CF gut may also be a key factor, as SCFAs exhibit tumour-suppressive properties [146,147]. For example, in an animal model of colitis-associated colorectal cancer, SCFAs have been shown to mediate reductions in proinflammatory cytokine release and tumour size and incidence [148]. All in all, while intestinal dysbiosis and inflammation tend to be clinically silent in CF, they may be associated with serious complications. This emphasises the importance of optimising gut health in the management of CF.

## 8. Microbiome Modulation with Probiotics

Improving gut health through microbiome modulation is gaining traction in GI and respiratory diseases [149,150]. The microbiome can be modulated through administration of a single or combination of commensal strains (probiotics), indigestible carbohydrates to promote the expansion of commensal strains (prebiotics), or a combination of both (synbionts). Probiotics were first described in 1907, and have been utilised as a beneficial dietary supplement since. In 2002, a consensus was reached by a joint FAO/WHO working group on the definition of probiotics: “Live microorganisms that, when administered in adequate amounts, confer a health benefit on the host” [151]. Existing probiotic preparations are based primarily on strains from the genus lactobacilli, bifidobacterial, and other lactic acid-producing bacteria (LAB) isolated from fermented dairy products and faecal microbiome samples [152]. However, rapidly expanding research into host–microbe interactions is increasing the impetus for the development of next-generation probiotics from beneficial microbes including *Akkermansia, Eubacterium, Propionibacterium, Faecalibacterium,* and *Roseburia* species [153,154,155].

Probiotics have reported beneficial effects in diseases with links to a GI dysbiosis, inflammation, and respiratory function [15,152,156]. However, knowledge gaps exist related to robust evidence-based probiotic use as a result of the significant heterogeneity between studies and variability in the probiotic strains studied. Specific probiotic strains have been indicated in the reduction in necrotizing enterocolitis (NEC) incidence [157] and the management of *Clostridium difficile* [158,159], though the quality of evidence remains low [160]. The rise of in vitro, animal, and cell culture research has expanded our understanding of prosed mechanisms of action, and include direct interaction with commensal gut microbiota, modulation of the immune system, production of organic acids, colonization resistance, improved barrier function, production of hormones and other small molecules with systemic effects, and probiotic–host interactions mediated by cell surface structures [149].

### 8.1. Mechanisms of Action

*Interaction with microbiome.* The direct interaction with the microbiome is mediated through the increasing microbial stability [161,162,163], cross-feeding [164], substrate formation, and antagonistic action through competition and production of antimicrobials and bacteriocins [165,166,167]. Competitive exclusion and inhibition of pathogenic species is a primary function of probiotics. In 2007, Collado et al. [168] tested 12 probiotic strains against 8 pathogenic strains in a pig intestinal mucosa model, and found that all probiotic strains tested were able to inhibit and displace pathogenic species of *Bacterioides*, *Clostridium*, *Staphylococcus* and *Enterobacter*. Another in vitro assay demonstrated that *B. animalis subsp.*
*lactis BB-12* and *Lactobacillus reuteri DSM 17938* inhibited the growth of pathogenic bacteria *E. coli* [169]. Likewise, *Lactobacillus paracasei FJ861111.1* has demonstrated significant inhibition against several common intestinal pathogens including *Shigella dysenteriae,*
*Escherichia coli,* and *Candida albicans* [159].

*Modulate immune system.* The interaction between microbiota and the immune system, reviewed in [170], has impacts systemwide. Probiotics have been shown to modulate immune function through an increase in anti-inflammatory cytokines [171,172], a reduction in proinflammatory cytokines [149,152,173,174], and augmentation of vaccines and antibody response [175,176,177]. The most common species to demonstrate immune modulation include *Lactobacillus*, *Bacillus*, and *Bifidobacterium*, and the genus *Saccharomyces* [178]. The modulation of the immune system through probiotics is not consistent across species or strains, and exhibits variability between hosts [149,172]. Yet recently, Sanders et al. [179] identified that some immune modulatory mechanisms related to cell surface infrastructure were conserved across species and even genera.

*Production of organic acids.* Probiotic species belonging to the Lactobacillus and *Bifidobacterium* genera produce lactic and acetic acids as end products of carbohydrate metabolism. These organic acids can reduce colonic pH, discouraging the growth of pathogens. Fredua-Agyeman et al. demonstrated that commercial cocultures of *Bifidobacterium* and *Lactobacillus* strains inhibited *Clostridioides difficile* growth in a pH-dependent manner [180]. Through the process of cross-feeding commensal bacterial species such as *Faecalibacterium*, *Lactobacillus*, and *Bifodobacterium*, probiotics can also increase levels of beneficial short-chain fatty acids (SCFA), including butyrate. SCFA have demonstrated anti-inflammatory and antitumour properties [86,181,182].

*Improve barrier function.* Tight junctions are critical to epithelial cell function, preventing translocation of microbial species and proinflammatory metabolites [183]. Probiotic *Lactobacillus* and *Bifidobacterium* strains have been shown to increase the expression of tight junction proteins [184,185] and reduce the severity of acute gastroenteritis in children through fortification of tight junctions [186]. Several *Lactobacillus* probiotic strains have also demonstrated regulatory effects on the epithelial mucus layer [153,187,188,189]. The demonstrated upregulation of mucin production genes and enhanced mucin secretion improves barrier function, inhibiting pathogen binding to epithelial cells [185].

*Production of small molecules with systemic effects.* Probiotic strains have been implicated in the production of a range of small molecules and hormones that influence systemic function. Interestingly, these include neurotransmitters such as cortisol, serotonin, tryptamine, noradrenaline gamma-aminobutyric acid (GABA), and dopamine, highlighting the potential of probiotics to modulate the gut–brain axis [190,191]. A range of satiety hormones and enzymes that can aid digestion are also produced by some probiotic strains. For example, *Streptococcus thermophilus* can facilitate lactose digestion through the production of microbial β-galactosidase [192].

*Probiotic–host interactions mediated by cell surface structures.* The cell surface architecture of probiotic strains is critical to probiotic–host cell interactions. Many Gram-positive probiotic strains share cell surface macromolecules that mediate these interactions, including surface layer associated proteins (SLAPS), mucin-binding proteins (MUBs), fibronectin binding proteins, and pili that interact directly with the intestinal epithelium, mucus, and gastrointestinal mucosa receptors. These demonstrated interactions reviewed in [179] can improve host barrier integrity, intestinal motility, and binding to intestinal and vaginal cells.

### 8.2. Probiotics in CF

The use of probiotics in CF has been investigated in 17 clinical trials. Of those 17 trials, 12 were RCTs, with 8 trials including children and 4 trials including both children and adults [15]. The number of subjects within the RCTs ranged from 22 to 81, and the trial duration ranged from 1 month to 12 months. The probiotic formulations varied in dosage from 10^8^ CFU/day to 10^11^ CFU/day. Strain formulations with six of the trials utilised a single Lactobacillus strain *L rhamnosus GG* [16,19,193,194] or *L. reuteri* [18,20,195], two trials utilised multistrain formulations with fructooligosaccharides (FOS) [17,66], and three trials utilised a multistrain without FOS [18,21,196]. Results from individual trials have cited a reduction in in inflammation [16,17,18], nutritional status [19], and pulmonary health outcomes [20,21] (Table 1).

From 2016 to 2021, six systematic reviews have attempted to synthesise the expanding evidence for probiotics in CF [15,197,198,199,200,201]. The first review in 2016 [201] examined a total of nine trials with a total of 275 subjects, and found that probiotics were likely to decrease gut dysbiosis and improve gut maturity and function. In 2017, three more reviews were published that were broadened to include evidence on pulmonary exacerbations and quality-of-life indicators [197,199,200]. The latest and most comprehensive systematic review was based on data from the 12 RCTs only [15]. Combined data from four trials (225 participants) found that probiotics may reduce pulmonary exacerbations when administered over a four-to-12-month period mean difference (MD) of −0.32 episodes per participant (95% confidence interval (CI) −0.68 to 0.03; *p* = 0.07)). The 95% confidence intervals included the possibility of both an increased and deceased number of exacerbations. The combined data from four trials (177 participants) also indicated that probiotics may reduce faecal calprotectin, MD −47.4 µg/g (95% CI −93.28 to −1.54; *p* = 0.04). Due to (i) a high risk of bias due to selective reporting; (ii) a high risk of bias due to incomplete outcome data; and (iii) a lack of generalisability, the evidence for these results was evaluated as low certainty [15].

The results from other biomarkers and health outcomes including lung function (forced expiratory volume at one second (FEV_1_)% predicted) (five trials, 284 participants); duration of antibiotic therapy (two trials, 127 participants); hospitalisation rates (two trials, 115 participants); height, weight, or body mass index (two trials, 91 participants); and reported health-related quality of life scores (1 trial, 37 participants) did not demonstrate any difference between placebo and treatment groups, (all low-certainty evidence). Only two studies included a microbial analysis, and insufficient data was available to analyse in the systematic review. Likewise, there was insufficient evidence to evaluate gastrointestinal symptoms. The probiotics evaluated in the RCTs were associated with four adverse events, including vomiting, diarrhoea, and allergic reactions [15].

Results from individual trials and systematic reviews have consistently indicated that probiotics are likely to have beneficial effects in CF, especially for inflammation and pulmonary exacerbations. However, all systematic reviews cited a limited amount of low-quality data as a barrier to justifying the inclusion of probiotics in current CF treatment protocols [15,197,198,199,200,201]. Furthermore, a high variation between trial protocols, probiotic formulation, dose, duration of therapy, and clinical outcomes measured make predictions about effective strains and dosages and clinical translation difficult. To address data quality, Coffey et al. [15] recommended multicentre RCTs of at least 12 months duration to best assess the efficacy and safety of probiotics for children and adults with CF.

## 9. Microbiome Modulation with Prebiotics

Modulation of the microbiome can also be targeted through the administration of generally non-digestible compounds known as prebiotics. Prebiotics provide health benefits by promoting the proliferation of commensal gut species and subsequent production of beneficial metabolites [202]. Prebiotics were first defined in 1995, with an updated definition published in 2017 as “a substrate that is selectively utilized by host microorganisms conferring a health benefit” [203]. Prebiotics occur naturally in foods such as breads, cereals, onions, garlics, and artichokes [204] but are also available as dietary supplements. The most established and well documented prebiotics include inulin, fructo-oligosaccharides (FOS), oligofructose, galacto-oligosaccharides (GOS), and lactulose [205,206]. Other potential prebiotics with expanding evidence of effect include resistant starch, high amylose maize starch (HMAS), glucans, arabinoxylan oligosaccharides, xylooligosaccharides, soybean oligosaccharides, isomalto-oligosaccharides, and pectin [206,207].

### 9.1. Mechanisms of Action

The underlying hypothesis of prebiotics is that the additional fermentable substrates drive the proliferation of keystone commensal bacteria, and subsequently the production of beneficial metabolites such as SCFA [152]. Commensal bacteria and SCFA metabolites then directly and indirectly improve host health through colonisation inhibition, increased barrier integrity, and immune modulation [183]. Currently, the mechanisms of action postulated for prebiotics are primarily based on in vitro models. Validation of proposed mechanisms and demonstrated effects in human models is limited.

*Interaction and modulation of microbiome.* Prebiotics in the form of undigestible carbohydrates promote proliferation of beneficial bacteria. As mentioned above, subsequent health benefits are a direct result of increased beneficial bacteria such as colonisation inhibition, or an indirect result of increased beneficial metabolite production such as improved barrier function and immune modulation [183]. In a randomised, double-blind, placebo-controlled clinical trial, a prebiotic intervention with GOS reduced intestinal permeability in obese adults. The probiotic intervention of *Bifidobacterium adolescentis* also reduced intestinal permeability, but interestingly, no synergistic effect was observed when the two were combined [208].

*Defence against pathogens.* As with probiotics, the modulation of the microbiome through prebiotics results in the generation of organic acids, reducing luminal pH, which inhibits growth of pathogens. The increase in commensal species also reduces nutrient availability for invasive species as described above. There is also evidence to suggest that GOS prebiotics can directly interfere with E. coli adhesion to tissue culture cells [209].

*Metabolic effects.* The metabolic effects of prebiotics have been the subject of several meta-analyses [210,211,212]. The evidence suggests that GOS and inulin can reduce high sensitivity C-reactive protein, plasma cholesterol, triglycerides, and fasting plasma insulin associated with obesity and diabetes. The exact mechanisms of action, duration of effects, or results from long-term consumption has not yet been established [211].

*Immune modulation*. Prebiotic immune modulation is primarily activated through microbial fermentation and subsequent production of SCFA metabolites. However, some prebiotics have been demonstrated to bind directly to some immune cell receptors [183,213]. Immune modulation is not consistent between probiotic categories or conditions, or even within conditions. A RCT with 259 infants concluded that GOS and long-chain FOS administered in formula may regulate immune function in infants, with a 50% reduction in atopic dermatitis, wheezing, and urticaria to when compared to non-prebiotic formula-fed infants [214]. Yet, a subsequent multicentre RCT with 365 infants found that while GOS supplementation altered faecal frequency and consistency, there was no effect on incidence of infection or allergic manifestation during the first year of life [215]. Likewise, conflicting studies in elderly individuals have proposed prebiotic GOS supplementation may have either no effect on immune function [216], or may increase immune function through enhanced phagocytic activity and activity of natural killer cells [217,218].

### 9.2. Prebiotics in CF

Prebiotics have been combined with probiotics in synbiotic preparations for use in CF probiotic trials [17,66,219]. However, only one study has investigated the exclusive use of prebiotics in CF for GI microbiome modulation [22]. Effective clinical use of prebiotics assumes a selective utilisation of the supplemented substrate by the recipient’s microbiota. It is not yet known if the disrupted microbiota in CF that is depleted in key SCFA-producing organisms, has the functional capacity to utilise prebiotic substrates. In a pilot study, Wang et al. [22] used a combination of metagenomic sequencing, invitro fermentation, amplicon sequencing, and metabolomics to investigate the HAMS fermentation capacity of 19 adults with CF and 16 non-CF controls. They demonstrated that despite low abundances of common taxa attributed to fermentation of HMAS (*Faecalibacterium, Roseburia* and *Coprococcus*), the production of butyrate and propionate was consistent with healthy control slurries, while the production of acetate was reduced. In the absence of *Faecalibacterium*, the CF SCFA biosynthesis was attributed to *Clostridium ss1*. Importantly, in a subset of CF patients, the presence of HAMS led to enterococcal overgrowth and the accumulation of lactate [22]. Likewise, a murine study found that supplementation with purified prebiotics inulin, fructooligosaccharides, or pectin may result in hepatocellular carcinoma in mice with pre-existing perturbed microbial communities [220]. These results demonstrated the potential for variable responses to prebiotics, dependent on the underlying microbiome [220].

In the absence of further CF-specific research, we examined evidence of prebiotics in GI disorders with an inflammatory link, including ulcerative colitis (UC), Crohn’s disease, colorectal cancer (CRC), and chronic respiratory disease including *Psuedomonas aeruginosa* infections, asthma, and emphysema (Table 2). In colitis animal models, a range of prebiotics including 2-fructosyl lactose [221], barley leaf insoluble dietary fibre (BLIDF) [222], psyllium [223], wheat bran [224], and butyrate [225] have been shown to reduce colitis symptoms and inflammatory markers, while increasing bacterial diversity and SCFA levels. Specific inflammatory markers, bacterial species, and SCFA vary between studies and prebiotic type (Table 2). The effects of inulin-type fructins (ITF) at 7.5 g/day (*n* = 12) or 15 g/day (*n* = 13) were tested in a human trial with 25 patients with mild/moderate UC [226]. The high-ITF-dose group showed significantly reduced colitis and calprotectin concentrations, and increased butyrate levels. The bacterial species *Bifidobacteriaceae* and *Lachnospiraceae* also increased in the high-dose group, but their abundance was not correlated to improved disease scores. The lack of taxonomic correlation suggested that functional shifts may be more relevant than compositional shifts in UC (Table 2). A RCT involving 140 preoperative patients with CRC investigated the role of prebiotics (fructooligosaccharides, xylooligosaccharides, polydextrose, and resistant dextrin) on immune function and intestinal microbiota. They reported that probiotics led to improved serum immunological markers and abundances of commensal bacterial species before surgeries [227]. Prebiotics did not protect from surgery-related microbial stress observed in both postoperative groups (Table 2).

In 2015, a mouse model study investigated the effects of dietary-pectin-derived acidic oligosaccharides (pAOS) on *Pseudomonas (P) aeruginosa,* and found that pAOS may limit the number and severity of pulmonary exacerbations chronically infected with *P. aeruginosa* [232] (Table 2). These results are highly relevant to individuals with CF, but have not been validated in human trials. In another small RCT study, either a placebo or a galactooligosaccharides (GOS) preparation highly selective to Bifidobacterium (B-GOS) was given to 10 adults with asthma-associated, hyperpnoea-induced bronchoconstriction (HIB) and 8 adult controls. Pulmonary function remained unchanged in the control group, but in the HIB group, FEV1 was attenuated by 40%, and the baseline chemokine CC ligand and TNF-a was reduced after B-GOS supplementation [237] (Table 2). As with probiotics, the large number and source of prebiotics limits the ability to effectively compare treatments across studies and conditions.

### 9.3. Microbiome Modulation with Diet

There is an increasingly apparent link between diet, microbiome modulation, and host health [238,239]. In the first few years of life, the interactions between diet and the microbiome are especially relevant. The oligosaccharides present in breast milk encourage the colonization of *Bifidobacterium* spp. The subsequent metabolites produced by *Bifidobacterium* spp. support the expansion of the microbiome through cross-feeding, and promote immune tolerance to other commensal bacteria [240,241]. The progression of the microbiome develops alongside increases in diet diversity, with the introduction of solid food triggering a rapid expansion in the bacterial community and the subsequent quantity and variety of metabolites. Metabolites associated with solid food ingestion, primarily butyrate, have been demonstrated to drive the maturation of the mucosal barrier in Caco-2 cells [242], which is critical to colonization inhibition of pathogens [243]. As mentioned earlier, iatrogenic factors such as antibiotic use can disrupt microbiome progression [41]. More broadly, early life microbiome disruption has been implicated in the development of autoimmune disease in mouse models [244], and has been associated with lasting metabolic and autoimmune disease consequences in observational studies [245,246,247,248].

The progression of the microbiome continues until it establishes near-adult levels of diversity at 3–5 years of age. Once established, the microbiome is more resistant to disruption [249]. However, diet remains a key modulator, with accumulating evidence of the role of dietary components in local inflammation, intestinal barrier function, and host immune dysregulation [220]. High-fat diets promote the translocation of certain bacteria by enhancing intestinal permeability and preferentially increasing the relative abundances of lipopolysaccharide-bearing bacteria [82,83,250,251,252,253]. Dietary fibres microbially fermented in the gut lead to the production of short-chain fatty acid metabolites, which have demonstrated roles in immune regulation [254,255,256], maintenance of epithelial barrier [223,257], and microbiome modification [258]. Low-fibre diets also have long-term implications for cancer [259] and metabolic and autoimmune diseases [260,261].

### 9.4. Diet in CF

Consistent with the general population, early-life microbiome development in CF is correlated to infant breastfeeding [262] and the initiation of solid foods [133]. Infant breastfeeding and solid food intake are further linked to the respiratory microbiome [262] and health outcome indicators, supporting the concept of a gut–lung axis in CF [133]. The progression of the CF microbiome is disrupted in early life [53,54,59], characterized by reduced taxonomic and functional profiles (dysbiosis) [7] that persist into adulthood [12]. No dietary fibre interventions have been trialled in CF, but epidemiological studies and preclinical and clinical trials support the potential of fibre to modulate the microbiome structure [234] and improve function in chronic gastrointestinal [263] and respiratory conditions [235,264].

A low-fat, high-fibre diet in individuals with ulcerative colitis improved the quality of life (QoL) as quantified through the QoL IBD survey. The intervention led to increased acetate and tryptophan levels and modulated the microbiome, increasing the abundance of *Bacteroidetes* and *Faecalibacterium* [228] (Table 2). A high-fibre diet also reduced inflammation and attenuated pathological changes associated with emphysema in an emphysema mouse model exposed to cigarette smoke. Alveolar destruction and inflammatory cytokines in bronchoalveolar lavage fluid (BALF) were reduced, while SCFA were increased [235] (Table 2). In contrast, saturated fat decreased microbial diversity and increased colitis severity and the TH1 mucosal response in a DSS-induced colitis mouse model [236] (Table 2). In 37 adults with asthma, high-fat intake was also demonstrated to increase inflammation and attenuate both the duration and magnitude of recovery from an aerosol-administered bronchodilator. However, the potential role of the microbiome was not investigated [265]. A range of micronutrients and antioxidant supplements have been trialled in CF to ameliorate fat-soluble vitamin deficiencies, altered fatty-acid synthesis, and increased oxidative stress [266]. In 2020, a meta-analysis concluded that the benefits sometimes observed across 8 antioxidant and 15 essential-fatty-acid supplementation studies was not consistent enough to recommend their routine use in CF [266]. With the exception of an exploratory study correlating vitamin D insufficiency with increases in potential pathogenic species [267], the effect of these supplementations on the microbiome have not been widely explored.

## 10. Considerations for Clinical Application and Future Studies

A survey from a CF clinic in the USA found that 60% of CF patients currently use some type of probiotic [268], despite the majority of probiotics available for sale having little to no evidence to support effectiveness, dose, or disease specificity [269]. Conditional use of specific probiotic strains in (non-CF) gastrointestinal disorders has been recommended for prevention of antibiotic-associated *C. difficile* infections, pouchitis, and prevention of necrotizing enterocolitis [160]. While these conditions may impact on individuals with CF, recommendations have not been validated specifically in CF. Evidence from CF-specific RCTs suggest probiotics are likely to improve GI inflammation and reduce the incidence of pulmonary exacerbations. Yet, the highly variable, low-quality data has been insufficient to determine ideal strains, dosage, and treatment duration, constraining the implementation of probiotics into routine CF care.

The evidence for prebiotics lags behind that for probiotics. There was only one clinical trial for the use of prebiotics in CF, and the results were mixed. Some individuals demonstrated successful microbial modulation and an increased production of butyrate and propionate. However, a subset of CF reactions exhibited enterococcal overgrowth, resulting in lactate accumulation and reduced SCFA biosynthesis [22]. The altered microbiome and predisposition to pathogenic overgrowth in CF highlights the need for standardised preparations of well-characterised prebiotics to be investigated specifically within the CF population to adequately evaluate safety and efficacy.

The increased energy needs of individuals with CF are currently being meet through “energy-dense, nutrient-poor” diets with excess saturated fats and inadequate fibre intakes [113]. As previously discussed, high-fibre, low-fat diets are associated with improved inflammation, immune modulation, and gut barrier function outcomes. While simply adding more dietary fibre to existing CF diets is tantalizing, interventional dietary trials have demonstrated that increasing dietary fibre intake does not necessarily translate to increased SCFA production or improvements in disease outcomes [220,270]. This is pertinent to members of the CF population, who are likely to have altered SCFA-generating pathways [7,56,60]. High-quality randomized trials with well-defined dietary components are essential to provide justification for modulating microbiome–host effects through diet. Considering the uptake of highly effective modulator therapies, a re-evaluation of dietary recommendations with a focus on diet quality and individual energy requirements is also recommended.

As outlined in previous reviews, there is a need for large, well-designed, longitudinal and multicentred clinical trials to effectively evaluate the safety and efficacy of probiotics in CF. While this is also true for prebiotics, there is a paucity of prebiotics research in CF and a plethora of substrates with prebiotic potential. The use of organoids, cell lines, and or animal models may be an economic option to demonstrate beneficial effects and mechanisms of action across a variety of compounds before proceeding to clinical trials. Likewise, there are a variety of dietary interventions and supplements that may have beneficial host–microbiome effects. High-quality, randomized studies with well-defined compounds are needed to evaluate safety and efficacy in CF before dietary interventions or supplements can be utilised to modulate the CF microbiome.

## 11. Conclusions

Although the exact mechanisms are not yet fully elucidated, the host–microbiome interactions in CF are critical to the incidence of GI inflammation and disease.

Targeted diet-based therapies provide an opportunity to modulate the altered CF microbiome to counter the early disruption to microbiome progression, and could transform GI and respiratory disease outcomes. The expansion of metagenomic, proteomic, and transcriptomic analyses continues to illuminate CF specific taxonomic and functional alterations. This knowledge will be critical to the development of next-generation precision probiotics and prebiotics. For now, there is insufficient evidence to support the safe and effective use of prebiotics in CF, but probiotics and a re-evaluation of the CF diet may be beneficial. Critically, there is a need for personalised approaches that understand an individual’s baseline microbiota and can manage potential microbiome-modulating therapies alongside existing medications and complex nutritional needs.

## Figures and Tables

**Figure 1 nutrients-14-00480-f001:**
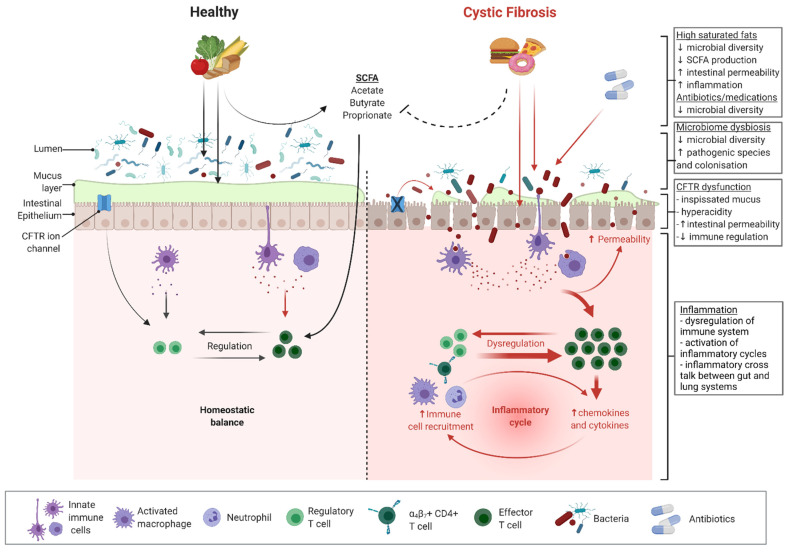
Microbiome- and CFTR-related dysfunction and inflammation in cystic fibrosis. Black arrows indicate direction of known homeostatic effects. Red arrows indicate direction of known inflammatory effects. Broken lines indicate proposed mechanisms of inhibition or dysfunction. Figure was created with Biorender.com.

**Table 1 nutrients-14-00480-t001:** Evidence for the use of probiotics in CF from RCTs.

Year	Probiotic Preparation (Dose)	Study Design	Duration	Probiotic Participants	Primary Results	Ref
1998	*L. rhamnosus strain GG* (6 × 10^9^ CFU/day)	RCT (Cross-over)	6 months	28	Increased weight gain (placebo 2.7 ± 2.5%, probiotic 8.7 ± 8.1%, *p* < 0.05). Reduced risk of infections (infections requiring antibiotic treatment per child in 6 months, placebo 39 and 1.7 ± 0.3, probiotic 19 or 0.9 ± 0.6, (*p* < 0.05)). Reduction in abdominal pain (placebo = 6 patients with abdominal pain, probiotic = 1, *p* < 0.05).	[194]
2007	Lactobacillus GG (6 × 10^9^ CFU/day)	RCT (Cross-over)	6 months	38	Reduction in pulmonary exacerbations (median 1 vs. 2, range 4 vs. 4, median difference 1, CI 95% 0.5–1.5; *p* = 0.003). Reduction in hospital admissions (median 0 vs. 1, range 3 vs. 2, median difference 1, CI 95% 1.0–1.5; *p* = 0.001). Increase in FEV1 (3.6% ± 5.2 vs. 0.9% ± 5; *p* = 0.02) and body weight (1.5 kg ± 1.8 vs. 0.7 kg ± 1.8; *p* = 0.02).	[19]
2009	CasenBiotic ^a^ (1 × 10^8^ CFU/day) VLS3 ^b^ (9 × 10^11^/day)	RCT (Cross-over)	6 months	40	Increased Quality of Life score from the PedsQL^TM^ survey. (Probiotics group—parent-reported, 0.87 higher (SD 0.19 higher to 1.55 higher)), (Probiotics group—child-reported, 0.59 higher (SD 0.07 lower to 1.26 higher)).	[18]
2013	Protexin capsule ^c^ (2 × 10^9^ CFU/day)	RCT (Parallel)	1 month	20	Rate of pulmonary exacerbation significantly reduced among probiotic group (*p* < 0.01). Parent-reported quality of life improved in probiotic group compared with placebo group at 3rd month (*p* = 0.01), not significant at 6th month of probiotic treatment.	[21]
2013	Protexin Restor sachet ^d^ (1 × 10^9^ CFU/day)	RCT (Parallel)	1 month	24	Mean faecal calprotectin levels decreased with probiotics 56.2 µg/g, compared to placebo 182.1 µg/g (*p* = 0.031).	[17]
2014	*L. reuteri* DSM 17938 (1 × 10^8^ CFU/day)	RCT (Cross-over)	6 months	30	Significant improvement in gastrointestinal health (GIQLY score placebo 11.2 ± 0.3, probiotic 11.4 ± 0.3, (*p* = 0.0036)). Decreased calprotectin (μg/ml) (placebo 33.8 ± 23.5, probiotic 20.3 ± 19.3, (*p* =0.003)).	[195]
2014	*L. reuteri* ATCC55730 (10^10^ CFU/day)	RCT (Parallel)	6 months	30	Reduced pulmonary exacerbations (odds ratio 0.06 ([95% confidence interval (CI) 0–0.40); number needed to treat 3 (95% CI 2–7), *p* < 0.01). Reduced number of upper respiratory tract infections (odds ratio 0.14 ([95% CI 0–0.96); number needed to treat 6 (95% CI 3–102), *p* < 0.05).	[20]
2014	Lactobacillus GG (6 × 10^9^ CFU/day)	RCT (Parallel)	1 month	10	Reduced calprotectin concentrations from baseline, compared to placebo (164 ± 70 vs. 78 ± 54 µg/g, *p* < 0.05; 251 ± 174 vs. 176 ± 125 µg/g, *p* = 0.3).	[16]
2018	Lactobacillus GG (6 × 10^9^ CFU/day)	RCT (Parallel)	12 months	41	No significant difference in odds of pulmonary exacerbations (OR 0.83; 95% CI 0.38 to 1.82, *p* = 0.643). No significant difference in odds of hospitalisations (OR 1.67; 95% CI 0.75 to 3.72, *p* = 0.211). No significant difference was for body mass index and FEV1.	[193]
2018	FOS + multi strain powder ^e^ (10^8^–10^9^ CFU/day each strain)	RCT (Parallel)	90 days	22	No significance difference in FEV1 and nutritional status markers. Patients with *Staphylococcus aureus* + supplementation had reduced NOx (*p* = 0.030), IL-6 (*p* = 0.033), and IL-8 (*p* = 0.009).	[66]
2018	*L. rhamnosus* SP1 (DSM 21690) and *B. animalis lactis* spp. BLC1 (LMG 23512) (10^10^ CFU/day)	RCT (Cross-over)	4 months	31	No significant changes in the clinical parameters (BMI, FEV1%, abdominal pain, exacerbations). Normalization of gut permeability was observed in 13% of patients during probiotic treatment.	[196]

^a^ CasenBiotic (CasenFleet) 100 million (108 CFU/day), *L. reuteri Protectis* (DSM 17938), sweeteners (isomaltose (E-953), xylitol (E-967)), calcium stearate, palmitic acid, citric acid, strawberry aroma as a capsule. ^b^ VLS3 (Faes Farma) 450 million, *B. breve*, *B. longum*, *B. infantis*, *L. acidophilus*, *L*. *plantarum*, *L. paracasei*, *L. delbrueckii subsp. bulgaricus*, *S. thermophilus* as a powder sachet. ^c^ Protexin capsule containing *L. casei, L. rhamnosus, S. thermophilus, B. breve, L. acidophilus, B. infantis*, and *L. bulgaricus*. ^d^ Protexin Restor sachet, FOS and a mixture of 1 × 10^9^ CFU/sachet bacteria (*L. casei, L. rhamnosus, S. thermophilus, B. breve, L. acidophilus, B. infantis, L. bulgaricus*). ^e^ FOS + multistrain powder (5.5 g), *L. paracasei*, *L. rhamnosus, L. acidophilus*, and *B. lactis.*

**Table 2 nutrients-14-00480-t002:** Evidence of prebiotics and dietary effects on microbiome in chronic inflammatory and respiratory disease.

Dietary Component	Study Model	Disease Type	Effect on Disease	Effect on Gut Microbiome	Effect on Host Biomarkers	Ref
Specific diet						
Low fat, high fibre	Human	Ulcerative colitis (IBD)	↑ QoL IBD questionnaire scores	↑ *Bacteroidetes, Faecalibacterium prausnitzii* ↓ *Actinobacteria*	↑ Acetate, tryptophan ↓ Lauric acid	[228]
Monosaccharides						
High-sugar diet	Mouse	DSS-induced colitis (IBD)	↑ Colitis	↑ *Verruncomicrobiaceae, Porphyromonadaceae* ↓ α-Diversity, *Prevotellaceae, Lachnospiraceae, Anaeroplasmataceae*	↑ Intestinal permeability, proinflammatory cytokines, BMDM reactivity to LPS.	[229]
Artificial sweetener	Mouse	SAMP1/YitFc ileitis (Crohn’s disease)	No change	↑ *Proteobacteria*	↑ Ileal myeloperoxidase reactivity	[230]
Milk oligosaccharides						
GOS	Human crossover	NA	NA	↑ *Bifidobacterium* ↓ *Ruminococcus, Synergistes, Dehalobacterium, Holdemania*	↓ Butyrate (NS), *Bacteroides* predicts OGTT	[231]
pAOS	Mouse	*P. aeruginosa* infection	↑ Bacterial clearance	↑ *Bifidobacterium, Sutturella wadsworthia, Clostridiumcluster XI*	↑ Butyrate, propionate ↑ IFN-γ, t-bet gene, M1 macrophage, IL10 ↓ TNF a, IL-4, gata 3 gene	[232]
2′-Fucosyl lactose	Mouse	IBD	↓ Colitis	↑ *Ruminococcus gnavus* ↓ *Bacteroides acidifaciens, Bacteroides vulgatus*	↑ Acetate, propionate, valerate, TGFβ ↓ iNOS, IL-1β, IL-6	[221]
Plant polysaccharides						
Dietary fibre	Mouse	T-cell-transfer colitis (IBD)	↓ Colitis	No change in microbial load or *Clostridiales* abundance, metabolic changes between high-fibre and low-fibre diets presumed based on butyrate output	↑ Treg cells, caecal and luminal butyrate, Foxp3 histone H3 acetylation	[233]
Dietary fibre	Human, RCT meta-analysis	NA	NA	↑ *Bifidobacterium*, *Lactobacillus* No change in α-diversity	↑ Faecal butyrate FOS and GOS drove microbial shifts	[234]
Dietary fibre	Mouse	Emphysema	↓ Alveolar destruction and inflammation in BALF	↑ *Bacteroidetes* ↓ *Lactobacillaceae, Defluviitaleaceae*	↑ SCFA, bile acids, sphingolipids ↓ Macrophages and neutrophils in BALF ↓ mRNA expression of IFN-γ, IL-1β, IL-6, IL-8, IL-18, IRF-5, MMP-12, TNF-α, TGF-β, and cathepsin S	[235]
BLIDF	Mouse	DSS-induced acute colitis (IBD)	Reduced colitis symptoms	↓ *Akkermansia* ↑ *Parasutterella, Alistipes, Erysipelatoclostridium*	↑ SCFA, secondary bile acids, claudin-1 ↑ Occludin and mucin 2 expression	[222]
FOS	Human, crossover	NA	NA	↑ *Bifidobacterium* ↓ *Phascolarctobacterium, Enterobacter, Turicibacter, Coprococcus, Salmonella*	↓ Butyrate, Bacteroides predicts OGTT	[231]
FOS, XOS, polydextrose, resistant dextrin	Human, RCT	CRC	↓ Inflammation	(Preoperative) ↓ *Bacteroides* ↑ *Bifodobacterium, Enterococcus* (Postoperative) ↓ *Bacteroides* ↑ *Enterococcus, Lactococcus, Streptococcus*	(Preoperative) ↑ IgG, IgM, transferrin (Postoperative) ↑ IgG, IgA, suppressor/cytotoxic T cells CD3+CD8+, total B lymphocytes	[227]
ITF	Human, RCT	Ulcerative colitis	↑ Remission ↓ Colitis	↑ *Bifidobacteriaceae, Lachnospiraceae* (not correlated with colitis reduction)	↑ Total SCFA, butyrate ↓ Faecal calprotectin	[226]
Psyllium	Mouse	DSS-induced, T-cell-transfer colitis (IBD)	↓ Colitis	↑ α-Diversity ↓ Microbial density	↑ Butyrate, Treg cells ↓ IL-6, faecal LCN-2, intestinal permeability	[223]
Wheat bran	Pig	NA	↓ Inflammation pathways	↑ *Bifidobacterium, Lactobacillis* ↓ *Escherichia coli*	↓ TNF-α, IL-1β, IL-6 and TLRs/MyD88/NF-κB pathways	[224]
SCFA						
Butyrate	Mouse	IBD	↓ Colitis	↑ α-Diversity (NS), *Lactobacillaceae, Erysipelotrichaceae* ↓ IgA-coated bacteria, *Prevotellaceae*	↓ TNF, IL-6, infiltration of inflammatory cells in colonic mucosa, acetate	[225]
Dietary fats						
Saturated fats	Mouse	Il10−/−, DSS-induced colitis (IBD)	↑ Colitis	↑ *Bacteroidetes, Bilophila wadsworthia* ↓ α-Diversity, *Firmicutes*	↑ TH1 mucosal response due to change in bile acid production	[236]

↑, increased; ↓, decreased; BALF, bronchoalveolar lavage fluid; BMDM, bone-marrow-derived macrophages; Bregs, regulatory B cells; CRP, C-reactive protein; DSS, dextran–sulfate–sodium; FOS, fructooligosaccharides; GOS, galactooligosaccharides; IBD, inflammatory bowel disease; ILA, indole-3-lactic acid; iNOS, inducible nitric oxide synthase; ITF, inulin-type fructans; LCN-2, lipocalin-2; LPS, lipopolysaccharide; NA, not applicable; NS, not significant; RCT, randomized controlled trial; SCFA, short-chain fatty acids; TGFβ, transforming growth factor-β; TH, T helper; TLR, Toll-like receptor; TNF, tumour necrosis factor; QoL, Quality of life; OGTT, oral glucose tolerance test; Treg, regulatory.

## Data Availability

Not applicable.

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
