# Peer review of "What Do We Know about the Microbiome in Cystic Fibrosis? Is There a Role for Probiotics and Prebiotics?"

_nutrients, 2022, doi:10.3390/nu14030480_

Round 1

Reviewer 1 Report

This is a very good, comprehensive review on i) microbiota dysbiosis and inflammation in cystic fibrosis and ii) on the gut microbiome modulation with diet, probiotics or prebiotics.

In the second part, several useful tables summarize the clinical trials data.

In the first part, scheme(s) and/or graphical abstract(s) illustrating / summarizing the main microbiota disorders linked to CF, their potential causes and consequences (interaction with the immune system, inflammation, epithelial barrier functions, potential clinical significance…) would help the reader.

Author Response

This is a very good, comprehensive review on i) microbiota dysbiosis and inflammation in cystic fibrosis and ii) on the gut microbiome modulation with diet, probiotics or prebiotics.

Thank you for your comments

In the second part, several useful tables summarize the clinical trials data.

In the first part, scheme(s) and/or graphical abstract(s) illustrating / summarizing the main microbiota disorders linked to CF, their potential causes and consequences (interaction with the immune system, inflammation, epithelial barrier functions, potential clinical significance…) would help the reader.

Thank you for your suggestion. We have now added a figure that summarises the microbiome and CFTR related dysfunction in CF. 

Reviewer 2 Report

The manuscript by Josie Maree van Dorst et al. discussed about the microbiome in Cystic Fibrosis and potential applications of probiotics and prebiotics for modulating the dysbiosis of Cystic Fibrosis. The topic is of interest to the readers and I have the following suggestions: 

1, Could the authors add a figure to summarize the major differences of the microbiome in healthy controls and patients with Cystic Fibrosis? 

2, How dysbiosis and Cystic Fibrosis are connected? What are the mechanisms? I suggest the authors to add a figure to illustrate this point. 

3, For the future studies, the authors should give further thoughts and discuss  more. 

Author Response

The manuscript by Josie Maree van Dorst et al. discussed about the microbiome in Cystic Fibrosis and potential applications of probiotics and prebiotics for modulating the dysbiosis of Cystic Fibrosis. The topic is of interest to the readers and I have the following suggestions: 

1, Could the authors add a figure to summarize the major differences of the microbiome in healthy controls and patients with Cystic Fibrosis? 

We have now added a figure that summarises the differences between microbiome related dysfunction in CF compared to Healthy individuals. 

2, How dysbiosis and Cystic Fibrosis are connected? What are the mechanisms? I suggest the authors to add a figure to illustrate this point. 

We believe the figure mentioned above summarises the differences in microbiomes between CF and healthy individuals as well as the implications and interactions between dysbiosis and CF.  

3, For the future studies, the authors should give further thoughts and discuss  more. 

Thank you for this suggestion, we have added an extra paragraph in section 10 to address considerations and further thoughts on future studies.  

Round 2

Reviewer 2 Report

The authors have revised the manuscript.